# Human Cytomegalovirus Infection Induces High Expression of Prolactin and Prolactin Receptors in Ovarian Cancer

**DOI:** 10.3390/biology9030044

**Published:** 2020-02-27

**Authors:** Afsar Rahbar, Amira AlKharusi, Helena Costa, Mattia Russel Pantalone, Ourania N. Kostopoulou, Huanhuan L. Cui, Joseph Carlsson, Angelique Flöter Rådestad, Cecilia Söderberg-Naucler, Gunnar Norstedt

**Affiliations:** 1Department of Medicine, Solna, Division of Microbial Pathogenesis, BioClinicum, Karolinska Institutet, 171 64 Solna, Sweden; hmarscosta@gmail.com (H.C.); mattia.pantalone@ki.se (M.R.P.); ourania.kostopoulou@ki.se (O.N.K.); leah.cui@ki.se (H.L.C.); cecilia.naucler@ki.se (C.S.-N.); 2Division of Neurosurgery, Karolinska University Hospital, 171 64 Stockholm, Sweden; 3Department of Physiology, College of Medicine and Health Sciences, Sultan Qaboos University, Muscat 135, Oman; 4Division of Pathology and Cytology, Karolinska University Hospital, 171 77 Stockholm, Sweden; joseph.carlson@ki.se; 5Department of Oncology and Pathology, BioClinicum, Karolinska Institutet, 171 64 Solna, Sweden; 6Department of Women’s and Children’s Health, Karolinska Institutet, 171 77 Stockholm, Sweden; angelique.floter-radestad@ki.se; 7Division of Obstetrics and Gynecology, Karolinska University Hospital, 171 77 Stockholm, Sweden; 8Department of Biochemistry, College of Medicine and Health Sciences, Sultan Qaboos University, Muscat 135, Oman; gunnar.norstedt@ki.se

**Keywords:** prolactin, prolactin receptor, HCMV, ovarian cancer

## Abstract

One of the potential biomarkers for ovarian cancer patients is high serum level of prolactin (PRL), which is a growth factor that may promote tumor cell growth. The prolactin receptor (PRLR) and human cytomegalovirus (HCMV) proteins are frequently detected in ovarian tumor tissue specimens, but the potential impact of HCMV infection on the PRL system have so far not been investigated. In this study, HCMV’s effects on PRL and PRLR expression were assessed in infected ovarian cancer cells (SKOV3) by PCR and Western blot techniques. The levels of both PRL and PRLR transcripts as well as the corresponding proteins were highly increased in HCMV-infected SKOV3 cells. Tissue specimens obtained from 10 patients with ovarian cancer demonstrated high expression of PRLR, HCMV-IE, and pp65 proteins. Extensive expression of PRLR was detected in all examined ovarian tumor tissue specimens except for one from a patient who had focal expression of PRLR and this patient was HCMV-negative in her tumor. In conclusion, PRL and PRLR were induced to high levels in HCMV-infected ovarian cancer cells and PRLR expression was extensively detected in HCMV-infected ovarian tissue specimens. Highly induced PRL and PRLR by HCMV infection may be of relevance for the oncomodulatory role of this virus in ovarian cancer.

## 1. Introduction

Ovarian cancer is often diagnosed at an advanced stage and is a major cause of morbidity and mortality in women worldwide [1]. Despite advanced therapies, the five-year overall survival of these patients has not improved over many years. The underlying cause of ovarian cancer is not completely known. Only about 10–15% of ovarian tumors are linked to genetic susceptibility [2] including inherited gene mutations in breast cancer (BRCA) genes 1 and 2 and mutations associated with Lynch syndrome [3]. Other factors that have been suggested to be involved in the development of ovarian tumors include estrogen hormone replacement therapy, the age at onset of menstruation and menopause, hormonal status, infectious agents, and immunological factors [4].

During the past decades, a link between human cytomegalovirus (HCMV) and cancer has been proposed based on the discovery of a high prevalence of viral proteins and nucleic acids in tumor tissue specimens and evidence of oncomodulatory abilities conferred by this virus [5,6,7,8,9,10,11,12,13]. Infection by HCMV is common in the healthy general population. HCMV belongs to the Herpes viridae family and is the largest virus in this group of viruses [14,15]. HCMV can infect many different cell types in the human body including monocytes, macrophages, fibroblasts, and epithelial and endothelial cells [14,15]. After a primary infection, the virus remains dormant in myeloid lineage cells mainly in monocytes and in hematopoietic CD34+ stem cells [15]. It is very rare to find an active infection in blood or tissue specimens obtained from healthy subjects. Thus, this virus is considered to be silent in healthy individuals. HCMV can reactivate from latency when these cells differentiate into macrophages or dendritic cells and the virus can then spread to new tissues [16,17,18]. This may occur during immune activation and inflammation and likely occurs from time to time during life. The reactivated virus per se can also contribute to increased inflammation in the body by activating NF-Kappa B and by enhancing the levels of inflammatory factors such as interleukins (IL); IL-1beta, IL-6, IL-8, tumor necrosis factor (TNF)-alpha, interferon (INF)-gamma, macrophage inflammatory protein (MIP)-1alpha, MIP-1ß and regulated on activation of normal T cell expressed and secreted (RANTES) [19,20,21], 5-lipoxigenase (5LO), and cyclooxygenase-2 (COX2) [22,23,24,25], which further aggravate inflammatory processes and support virus replication. Through sophisticated strategies, HCMV simultaneously avoids detection and elimination by the immune system. For example, HCMV causes downregulation of expression of human leukocyte antigen (HLA) class I and II molecules, induces production of immunosuppressive cytokines such as tumor growth factor (TGF)-beta, IL-10, and a functional viral interleukin-10 homologue [26,27], and mediates inhibitory effects on natural killer (NK) cells and T cells that result in immunosuppression and immunomodulation. An interesting connection between HCMV and prolactin (PRL) is indicated through cyclophilin A and cyclosporine [28], an immunosuppressive drug used in treatment of transplant patients. Of note, a functional interaction between cyclophilin and the prolactin receptor (PRLR) has also been reported [29]. 

Although most studies suggest that HCMV negatively influences tumor cells and confers oncomodulatory effects, some viral strains may also be oncogenic. A clinical strain of HCMV with oncogenic transforming ability was recently isolated from a pregnant woman with an acute HCMV infection [30]. Infection of normal mammary epithelial cells (HMECs) with this clinical isolate, HCMV-DB, resulted in cellular transformation and led to tumor formation in a mouse model. The transformed cells harbored an HCMV-RNA signature with upregulated oncogenic pathways including inactivation of retinoblastoma (Rb) and p53 proteins, activation of telomerase, proto-oncogenes c-Myc, Ras, Akt, and STAT3, and upregulation of cyclin D1 and Ki67 [31]. The transformed cells exhibited a triple-negative phenotype representing the most aggressive breast tumor with no current effective treatment and very poor prognosis [30,31], and grew fast to establish large tumors in mice. An HCMV-RNA signature analogous to that of infected cells was detected in breast cancer biopsies, but not in healthy human breast tissues [30,31]. These data are supported by our observations demonstrating high levels of HCMV in triple-negative breast cancer [32]. These observations also confirm data by Rapp’s group published in the 1970s demonstrating tumor transformation in vitro and tumor development in a mouse model by infection of normal cells with a clinical strain isolated from a patient [33]. However, most other HCMV strains do not appear to cause oncogenic transformation of normal cells.

Ovarian cancer is a serious condition and patients are in high need of new treatment options. Expression of the PRLR that belongs to the cytokine receptor family is for unknown reasons high in ovarian cancer [34]. The PRLR ligand prolactin (PRL) is a hormone that exerts many functions in addition to its most established role in lactogenesis. It stimulates proliferation of different cell types in the body and has been linked to enhanced tumor cell growth [35,36,37]. Increased PRLR levels in tumors may therefore stimulate cancer cell growth via PRL [38,39]. Recent findings demonstrate that PRL production is not only restricted to the pituitary gland [40], which makes this theory interesting. In humans, there is a specific gene promoter allowing for PRL production in extra-pituitary locations [41], which implies that PRL not only functions as a regular hormone but potentially also as a local growth factor. Activation of the PRLR by PRL leads to activation of the JAK/STAT pathway [39,42]. Several publications suggest that activation of the PRL/PRLR signaling pathways is linked to cancer via activation of PI3K, AKT, and MAPK pathways, which are important in tumorigenesis [43,44,45]. These pathways are also known to be activated by HCMV infection. We have recently reported a frequent presence of HCMV proteins in serous ovarian cancer, and found an association between high virus activity in the tumor and poor patient survival rates [46,47]. We therefore hypothesized that HCMV may affect the PRL/PRLR axis in ovarian cancer.

In the present study, we aimed to investigate the possible interactions between HCMV infection and the PRL/PRLR axis by analyzing the effect of infection on the expression of PRL and PRLR following experimental HCMV infection of ovarian cancer cells in vitro and by analyzing ovarian cancer tissue specimens for HCMV and PRLR proteins using immunohistochemical staining. Our data indicate a previously not recognized role for HCMV to regulate the PRL/PRLR system that may be of significance for the oncomodulatory role of HCMV in ovarian cancer.

## 2. Results

### 2.1. Increased PRL Transcript Levels in HCMV-Infected SKOV3 Cells

Uninfected and HCMV-infected ovarian cancer cells (SKOV3 cells) were sampled at 1, 3, and 5 dpi for detection of PRL transcripts. The HCMV-IE transcript levels in HCMV-infected SKOV3 cells at 1, 3 and 5 dpi were equal to *Ct* = 19,5, 19,2 and 19,1, respectively, which was sufficient infection efficiency as compared to our earlier published data on HCMV-infected MRC-5 cells used as positive control [48]. Significantly increased levels of PRL transcripts were detected in HCMV-infected SKOV3 cells at 3 and 5 dpi, but not at 1 dpi (Figure 1A). PRL transcript levels were further increased in infected cells that were pretreated with PRL 3 h before infection. This effect was not observed in infected cells treated at the time of infection when compared to uninfected cells with similar PRL treatment conditions (Figure 1B,C). PRL transcript levels tended to be slightly higher in infected cells pretreated with PRL 3 h before infection compared with untreated infected cells *(*Figure 1C). In sharp contrast, PRL stimulation of uninfected cells did not change the low PRL expression levels in these cells (Figure 1B,C). 

### 2.2. Increased PRLR Transcript Levels in HCMV-Infected SKOV3 Cells

Next, we examined the effects of HCMV infection on PRLR transcript levels in SKOV3 cells at 1, 3, and 5 dpi. PRLR transcript levels were significantly increased in infected cells at 3 and 5 dpi, but not at 1 dpi (Figure 2A, with sufficient infection efficiency as in Section 2.1). PRL pretreatment of cells starting 3 h before or at the time of infection did not alter PRLR transcript levels (Figure 2B,C). However, PRL pretreatment of uninfected cells increased PRLR transcript levels at 5 days post-treatment (Figure 2C).

### 2.3. HCMV Induces PRL and PRLR Protein Expression in HCMV-Infected SKOV3 Cells

Untreated and PRL-treated uninfected and HCMV-infected SKOV3 cells were examined for protein levels of HCMV-IE (IE86 and IE72), PRL, and PRLR at 5 dpi. HCMV-IE protein levels did not change by PRL pretreatment of infected cells (Figure 3). PRL protein levels were highly increased in supernatants of HCMV-infected cells (in untreated and PRL-treated cells, Figure 3A).

Densitometry analyses normalized to β-actin showed that the level of PRLR protein was significantly increased in HCMV-infected cells treated with PRL compared with uninfected cells (Figure 3A,B). The protein levels of PRL increased in HCMV-infected SKOV3 cells both in the presence or absence of PRL stimulation as compared to uninfected cells (PRL-treated or untreated cells, Figure 3A,C). 

### 2.4. Expression of PRLR in HCMV-Infected Ovarian Cancer Tissues

Ovarian cancer tissue specimens from 10 patients were examined for expression of HCMV-IE, pp65, and PRLR (Figure 4 and Table 1). This cohort of ovarian cancer patients was previously investigated for HCMV protein expression and we reported an association with the clinical outcome of these patients with poor survival rates among patients with high activity of HCMV in their tumors [46,47]. As previously reported, we found that expression of HCMV-IE and pp65 proteins were detected at different levels in these tissue specimens. The expression of PRLR was very abundant in all ovarian cancer samples, except for one from a patient who was negative for HCMV and this patient only had focal expression of PRLR in her tumor specimen. Of note, a majority of HCMV-positive cells within the tumor tissue specimens were expressing PRLR. However, not all PRLR-expressing cells within the tumor tissues were positive for HCMV proteins, which indicates both a direct and potentially also indirect effect of HCMV on PRLR expression. We observed that very low levels of PRLR were expressed in an HCMV-negative human placenta tissue specimen (Figure 4C). 

Extensive expression (scores 3–4) of HCMV-IE and pp65 proteins was observed in 5/10 (50%) and 3/10 (30%) of patients, respectively (Table 1). All patients except for one had extensive protein expression levels of PRLR (scores 3 and 4, Table 1).

## 3. Discussion

In the present study, we found evidence that HCMV infection can enhance the expression of PRLR and induce the production of PRL in ovarian cancer cells. We also found an association between extensive PRLR levels and extensive HCMV protein expression levels in tissue specimens obtained from ovarian cancer patients. Interestingly, PRL treatment of HCMV-infected SKOV3 cells resulted in increased protein levels of PRLR. Furthermore, protein levels of PRL in the infected cells were increased both in untreated and PRL-treated cells. In uninfected cells, PRL treatment led to elevated PRLR transcript levels but not enhanced PRLR protein levels, which may be explained by an inhibitory post-translational mechanism to control PRLR protein translation in PRL-treated cells. 

During the past decades, an oncomodulatory role of HCMV has been investigated in many laboratories and current data suggest that the presence of HCMV in tumors may lead to more aggressive cellular behaviors and a poor response to conventional therapies. Here, we found that HCMV-IE and pp65 proteins were expressed at various levels in ovarian tissue specimens. Extensive expression of HCMV-IE and pp65 was detected in 50% and 30% of ovarian tumor tissues, respectively. One patient was negative for both HCMV proteins and another patient was negative for HCMV-IE proteins in her tumor specimen, but was positive for HCMV-pp65 proteins. Extensive expression of PRLR was detected in all examined ovarian tumor tissues except in the only patient who was negative for HCMV. This patient interestingly also survived the longest (68.5 months) among these patients. To our knowledge, this is the first report demonstrating an effect of HCMV on PRL and PRLR in ovarian cancer cells. HCMV-induced production of PRL and its receptor may promote tumor growth in autocrine and paracrine manners. Binding of PRL to PRLR leads to phosphorylation and activation of JAK-2, STAT-5/-3, PI3K/Akt, and MAPK, which result in cellular proliferation, differentiation, and inhibition of apoptosis. Activation of these pathways is a well-known phenomenon in HCMV-infected cells [44,49] and HCMV-induced function of the PRL/PRLR axis may therefore result in increased cellular proliferation and tumor progression via binding to the highly abundant PRLR in tumors. 

Many studies have suggested a role for PRL in different forms of cancers. A majority of such studies have concerned so-called hormone-sensitive cancers such as breast and prostate cancer. PRL was shown to promote tumorigenesis by activating the Ras oncogene and by inducing malignant transformation in ovarian epithelial cells and enhanced tumor growth in SCID mice carrying mutations in tumor suppressor genes [38]. PRL was also shown to induce IGF-I and -2 signaling [50] that in turn stimulated PRL gene expression by a Ras-dependent mechanism [51]. Ras activation can also stabilize the PRLR [52] and existence of HCMV and an extensive expression of PRLR may involve activation of Ras [53]. In ovarian tumors, PRLR activation may also result in further induction of inflammatory mediators such as NF-kappa B, which plays a central role in inflammation by regulation of genes encoding for proinflammatory cytokines, adhesion molecules, chemokines, growth factors, COX2, and inducible nitric oxide synthase (iNOS) [54,55,56]. Inflammation in the tumor microenvironment and production of inflammatory cytokines such as TNF-α, IL-1β, and IFNγ (that could also be induced by HCMV) may lead to further induction of PRLR [57]. Inflammation-induced HCMV and/or PRLR signaling could thereby also contribute to carcinogenesis. 

Findings of increased PRLR levels in tumors combined with a frequently observed elevation of serum PRL in cancer patients formed the basis for a clinical trial to suppress serum PRL with dopamine receptor agonists. However, such treatment failed to demonstrate significant effects on tumor growth [58]. Due to the more recent demonstration of PRL production outside the pituitary gland, attempts have been made to modulate tumor production of PRL by using PRLR antagonists (PRLRA). A PRLR blocking monoclonal antibody showed significant antitumor activity against MCF7 breast cancer xenografts [59]. Our previous study in glioblastoma demonstrated increased PRLR expression levels and indicated some efficacy to reduce tumor cell growth by using PRLRA in experimental systems [39]. These data support the idea that the PRL system is relevant in tumorigenesis and that agents blocking the receptor or drugs conjugated with PRLRA may be of relevance for future pharmaceutical development [60].

However, while the effects of the PRL/PRLR axis may affect tumor growth per se and be impacted by HCMV, it may not play an essential role for tumor progression in the context of HCMV-positive tumors, as this virus has many other mechanisms by which it can affect tumor growth in the case of an absence of PRL/PRLR activation. The same scenario may also exist for other signaling pathways relevant in cancer that are affected by HCMV: EGFR, PDGFR, FGF, ETBR, and IGF [61,62,63,64,65,66]. Loss of activation of one of these pathways may be compensated by activation of another. Still, the knowledge of specific receptors/signaling pathways of relevance for cancer is valuable in attempts to develop drugs blocking different pathways. Furthermore, insights into receptor internalization can provide drugs with an ability to deliver compounds more specifically to tumors as exemplified in cases where monoclonal antibodies have been conjugated to anticancer drugs [67]. An alternative therapeutic approach is to target viral components of tumors. In support of this concept, we previously reported a clinical impact of high HCMV activity in patients with glioblastoma and ovarian cancer [46,47,68,69,70]. Patients with serous ovarian cancer who had focal expression of HCMV proteins in their tumors lived 14 months longer compared to patients with extensive expression of HCMV proteins in their tumors. For glioblastoma patients, an enhanced survival of 20 months was observed in patients with low-grade (focal) infections compared with patients with high-grade extensive infection. Importantly, lowering virus activity with antiviral therapy in glioblastoma patients indicate improved survival rates [71]. Glioblastoma patients, who were treated with the anti-HCMV drug valganciclovir, enhanced their median overall survival from 13.1 to 24.1 months. If undergoing a more optimal treatment protocol, their median overall survival increased to 54.6 months. Moreover, patients who received HCMV-directed immunotherapy via dendritic cell vaccination also demonstrated highly improved survival rates. We therefore speculate that HCMV is a driving force for disease progression of glioblastoma via many different pathways including the PRL/PRLR axis. Under such circumstances, antiviral strategies lowering HCMV´s activity in the tumor may enhance the survival chances for glioblastoma patients by decreasing the activity of several important signaling pathways in cancer that could be impacted by HCMV. A randomized, double-blinded, placebo-controlled study is currently recruiting patients to assess whether valganciclovir can improve overall survival of glioblastoma patients. Antiviral strategies may therefore also be relevant to investigate for patients with ovarian cancer who lack effective treatment options and who are positive in their tumors for HCMV.

## 4. Materials and Methods

### 4.1. Cell Culture, HCMV Infection, and PRL Treatment

The ovarian cancer cell line SKOV3 was obtained from American Tissue Culture Collection (ATCC, Manassas, VA, USA). SKOV3 cells were cultivated in McCoy′s medium containing 2 mM glutamine containing 10% Fetal Bovine Serum (FBS) (Gibco, Waltham, MA, USA), 100 U/mL penicillin, and 100 μg/mL streptomycin (Gibco, USA) at 37 °C, 5% CO_2_. HCMV strain VR1814 was used to infect the cells with a multiplicity of infection of 5 (MOI) at 1 day post-infection (dpi), 3 dpi, and 5 dpi. In order to design a study resembling different scenarios occurring in vivo, the treatment of the cells with human recombinant PRL (200 ng/mL, generous gift from Novo Nordisk A/S Denmark) started three hours before infection/at the time of infection with HCMV at MOI 5 for 5 dpi, or left as uninfected. 

### 4.2. Western Blotting

SKOV3 cells were infected with HCMV strain VR1814 at MOI 5 for 5 dpi. Cells were left untreated or pretreated with human recombinant PRL (200 ng/mL) for 3 h before infection. Cells were lysed with RIPA buffer (150 mM sodium chloride, 1% NP-40, 0.5% sodium deoxycholate, 0.1% SDS, and 50 mM Tris, pH 8.0), and protein was quantified by BCA assay (Thermo Fisher Scientific, Waltham, MA, USA). Proteins (25 μg) were separated on a NuPAGE 4–12% Bis Tris gel (Thermo Fisher Scientific), transferred to a PVDF membrane (Millipore, Burlington, MA, USA), and detected with antibodies against HCMV-IE (Argene, Marcy-l’Étoile, France), PRLR (Abcam, Cambridge, UK), prolactin (R&D Systems, Minneapolis, MN, USA), and beta actin (Sigma Aldrich, St. Louis, MO, USA). For detection of secreted PRL proteins, cell culture media was collected and proteins were precipitated with trichloroacetic acid (TCA) method before electrophoresis.

### 4.3. RNA Extraction and Quantitative Real-Time PCR (qPCR)

SKOV3 cells were untreated or pretreated with PRL, left uninfected or infected with HCMV strain VR1814 at MOI of 5, and collected at 1 dpi, 3 dpi, and 5 dpi. RNeasy Mini Kit (Qiagen, Hilden, Germany) was used according to the manufacturer´s instructions to extract RNA from lysed cells. Random primers and the High-Capacity cDNA Reverse Transcription Kit (Applied Biosystems, USA) was used to synthesize cDNA from extracted RNA samples. TaqMan Fast Universal PCR Master Mix (Life Technologies, Carlsbad, CA, USA) was used to quantify gene expression levels of the samples. The following specific TaqMan probes were used as previously described [70] custom-made HCMV-IE forward primer, 5′-TGA CGA GGG CCC TTC CT-3′ and reverse primer, 3′-CCT TGG TCA CGG GTGTCT-5′; probes, FAM-AAG GTG CCA CGG CCC G-NFQ) (Life Technologies) [48,72], PRLR (Hs01061477 m1, Thermofisher Scientific), PRL (Hs00168730_m1, Thermofisher Scientific). The PCR was performed using a Fast Real-Time PCR system (7900HT, Applied Biosystems, Foster City, CA, USA). The endogenous control B2M (Life Technologies, Carlsbad, CA, USA) was used for normalization and relative expression was determined by the 2-ΔΔ*Ct* methods. Three separate experiments were performed.

### 4.4. Immunohistochemical Staining

To investigate the expression of HCMV-IE, pp65, and PRLR in human ovarian cancer tissue sections, available paraffin-embedded tissue sections were obtained from 10 patients with ovarian cancer who underwent surgery at Karolinska University Hospital during February 2010 and July 2012 (Table 2). Clinical diagnosis of these patients was confirmed by a pathologist (JC) at Karolinska hospital. HCMV-negative human placenta tissue section was used as control. Ethical permission was obtained from the Stockholm regional ethical committee and the regional ethical committee at the Karolinska Institutet (Dnr: 2008/628-31/2, Dnr: 01-420, Dnr: 2009/1412-31, Dnr: 2012/654-31, Dnr: 1989:262, Dnr: 510/00).

Working conditions were optimized for heat-induced antigen retrieval, after deparaffinization and rehydration of tissue sections as described before. For antigen retrieval, the slides were treated with citrate buffer, pH 6.0 (Biosite, Solihull, UK), in a pressure cooker for 20 min and rinsed in Tris-buffered saline containing 0.05% triton (TBST). Endogenous avidin and biotin were neutralized using the Avidin/Biotin Blocking Kit (Dako, Glostrup, Denmark) and endogenous peroxidase was blocked by 3% H2O2 (Histolab, Stockholm, Sweden) for 15 min at room temperature (RT) in the dark. Fc receptor was blocked by treating tissue sections with Fc receptor blocker for 30 min at 20 °C (Innovex Biosciences, Richmond, CA, USA). Then, tissue sections were incubated with targeted primary antibodies diluted in a common antibody diluting buffer (BioGenix, Houston, TX, USA) against HCMV-IE (Chemicon International, Temecula, CA,

USA, HCMV-pp65 (BioGenex), and PRLR (Invitrogen, USA) and incubated at 4 °C overnight followed by washing and incubation with biotinylated anti-mouse secondary antibody diluted according to the manufacturer’s instructions (BioGenix) for 45 min at room temperature and incubation with streptavidin-biotin-peroxidase complex (BioGenix, USA). Haematoxylin (Histolab) was used for counterstaining. As negative control, the primary antibody was omitted from the staining procedure. Negative controls were performed in parallel with all experiments. In addition, an HCMV-negative human placenta tissue specimen served as control. This tissue specimen was double-stained for HCMV-IE and PRLR using the same IHC protocol as described above, except for the use of a different secondary antibody, alkaline phosphatase anti-rabbit secondary antibody (Dako), and Fast Red chromogen (Histolab).

The expression of different HCMV proteins and PRLR was scored as described earlier [26]: negative (0% positive cells), focal expression: 1 (<24% positive cells) and 2 (>25–50% positive cells), extensive expression: 3 (>50–75% positive cells) and 4 (>75% positive cells). Two researchers independently collected staining results and the pathological diagnosis was reviewed by a pathologist (JC). Sections were scanned using Hamamatsu Nano Zoomer-XR Digital slide scanner and visualized using Nano Zoomer Digital Pathology (NDP) viewer software (U12388-01; NDP.view2 Viewing).

### 4.5. Statistical Analysis

Cell culture experiments were performed in duplicates or triplicates in at least three independent experiments. Statistical significance of the differences was evaluated using unpaired, two-tailed Student’s *t*-test, ANOVA test with post hoc analysis. The *p* value was calculated and considered statistically significant if *p* < 0.05.

## 5. Conclusions

In conclusion, the infectious activity of HCMV can stimulate activation of local growth stimulatory agents including the PRL/PRLR system. HCMV induces high levels of PRL and PRLR transcripts and proteins in HCMV-infected ovarian cancer cells. As extensive expression of PRLR was found in HCMV-infected ovarian tumors, and the PRL/PRLR axis can promote tumor growth, this may provide yet another mechanism by which HCMV can promote disease progression of cancer. Future studies devoted to blocking HCMV infection may offer new treatment possibilities for ovarian cancer patients.

## Figures and Tables

**Figure 1 biology-09-00044-f001:**
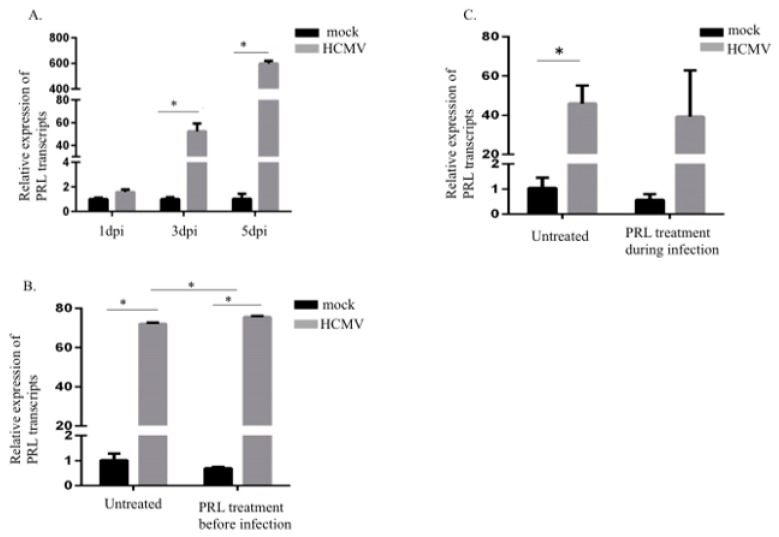
**Prolactin** (PRL) transcripts are increased in HCMV-infected SKOV3 cells. (**A**) Relative expression of PRL in SKOV3 cells infected with HCMV (MOI 5) compared to noninfected cells was determined by qPCR at 1, 3, and 5 dpi. (**B**,**C**) Relative expression of PRL in SKOV3 uninfected cells or infected with HCMV (MOI 5) after pretreatment or at the same time as treatment with PRL (200 ng/mL) was determined by qPCR at 3 dpi. Data is presented as mean ± SD (*n* = 3). Statistical significance is indicated as * *P* < 0.05.

**Figure 2 biology-09-00044-f002:**
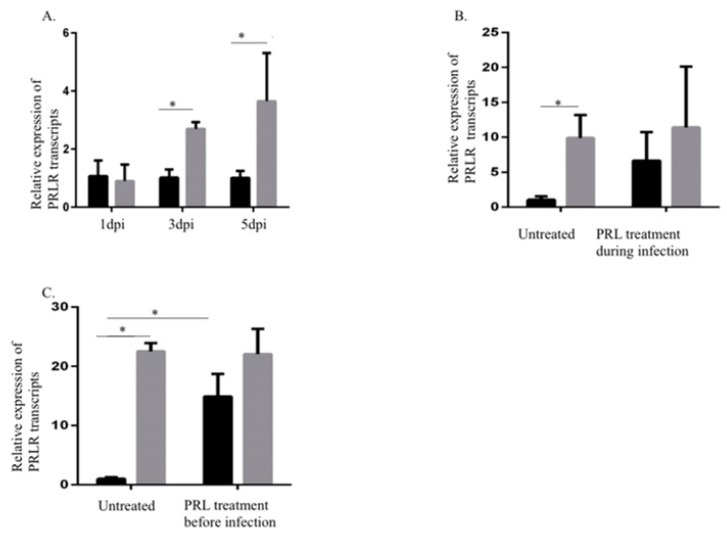
Increased prolactin receptor (PRLR) transcript levels in HCMV-infected SKOV3 cells. (**A**) Relative expression of PRLR in SKOV3 cells infected with HCMV (MOI 5) compared to noninfected cells was determined by qPCR at 1, 3, and 5 dpi. (**B**,**C**) Relative expression of PRLR in SKOV3 uninfected cells or infected with HCMV (MOI 5) after pretreatment or at the same time as treatment with PRL (200 ng/mL) was determined by qPCR at 3 dpi. Data is presented as mean ± SD (*n* = 3). Statistical significance is indicated as * *P* < 0.05.

**Figure 3 biology-09-00044-f003:**
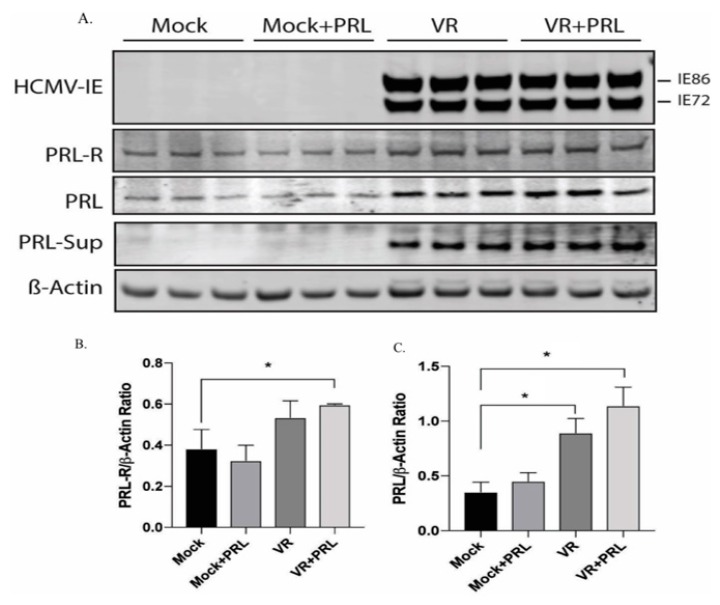
Increased protein levels of PRL and PRLR in HCMV-infected SKOV3 cells. HCMV-IE, PRLR, and PRL proteins were detected in untreated and PRL-treated (200 ng/mL) uninfected and HCMV-infected SKOV3 cells by Western blot assay. (**A**) PRL was detected in cell culture media derived from infected and uninfected cells after protein precipitation. (**A**–**C**) Both PRL and PRLR proteins were significantly increased in HCMV-infected cells treated with PRL. (**C**) PRL was detected in HCMV-infected cells without PRL treatment. Results are representative of three separate experiments. Data is presented as mean ± SD (*n* = 3). Statistical significance is indicated as * *P* < 0.05.

**Figure 4 biology-09-00044-f004:**
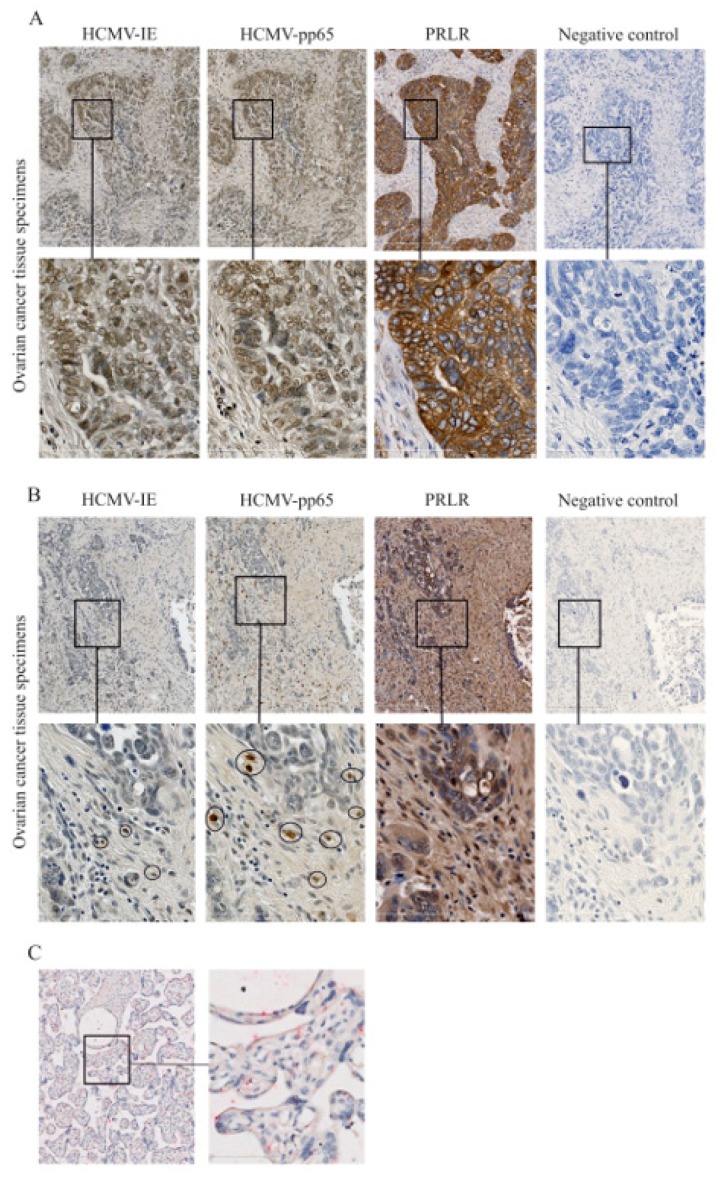
Expression of HCMV-IE, pp65, and prolactin receptor (PRLR) in ovarian tumor tissues. Expression of HCMV-IE, pp65, and PRLR proteins was monitored in ovarian tumor tissue specimens. (**A**) Extensive HCMV-IE, pp65, and PRLR were detected in a patient with serous adenocarcinoma and (**B**) focal expression of HCMV-IE and pp65 but extensive expression of PRLR were detected in another patient with serous adenocarcinoma. (**C**) HCMV-negative human placenta tissue was stained for HCMV-IE (brown color) and PRLR (red color). The lower row in the images shows the magnified area from images in the upper row. Scale bars: 50 µm.

**Table 1 biology-09-00044-t001:** Summarized data on expression of HCMV-IE, pp65, PRL, and PRLR in ovarian tumor tissues.

Patients	Ovarian Cancer	HCMV-IE	HCMV-pp65	PRLR	OS (m)
1	Other histotypes	0	0	1	68.5
2	Serous adenocarcinoma	4	3	4	40
3	Serous adenocarcinoma	0	1	3	42.5
4	Other histotypes	3	2	4	67
5	Other histotypes	4	4	4	41
6	Other histotypes	1	1	4	66
7	Serous adenocarcinoma	4	2	3	48
8	Serous adenocarcinoma	2	1	3	28
9	Serous adenocarcinoma	4	4	4	31
10	Serous adenocarcinoma	1	2	4	32

HCMV: human cytomegalovirus; IE: immediate early; PRLR: prolactin receptor; OS: overall survival; M: months.

**Table 2 biology-09-00044-t002:** Patient characteristics.

Patient Characteristics	
Ovarian Cancer (*n* = 10)	
Median age, years (range)	65 (42–74)
Stage	
IA, IC	*n* = 3
IIIC	*n* = 6
IV	*n* = 1
Median BMI (range)	26.5 (20–26.9)
Median CA125 level	
Initial	436 U/mL
After treatment	17.5 U/mL
Surgery	
R0	*n* = 6 (60%)
R1	*n* = 4 (40%)
Neoadjuvant chemotherapy before surgery	
Yes	*n* = 3 (30%)
No	*n* = 7 (70%)
Adjuvant chemotherapy after surgery	
Yes	*n* = 9 (90%)
No	*n* = 1 (10%)
Dead at study closure	*n* = 5 (50%)
Alive at study closure	*n* = 5 (50%)

BMI: body mass index; R0: complete cytoreduction (no visible tumor at the end of surgery); R1: resection (1–2 cm^2^ of tumor left at the end of surgery).

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
