# Peer review of "Human Cytomegalovirus Infection Induces High Expression of Prolactin and Prolactin Receptors in Ovarian Cancer"

_biology, 2020, doi:10.3390/biology9030044_

Round 1
Reviewer 1 Report
Reviewer comment
The present paper “Human Cytomegalovirus Infection Induces High Expression of Prolactin and Prolactin Receptors in Ovarian Cancer” submitted by Rahbar et al have demonstrated the high expression of HCMV induced prolactin and prolactin receptors in ovarian cancer. In this study, the authors used western blot, PCR other methods to delineate PRL and PRLR expression in the ovarian cancer cells as well as cancer tissues. Moreover, authors have also shown the presence of HCMV and increased level of PRL as well as PRLR in ovarian cancer tissues. These are the interesting findings and open a new possible role of HCMV in ovarian cancer. However, the author did not describe how HCMV infection leads to the high expression level of PRL and PRLR. It would be interesting to delineate the mechanism of how HCMV causing higher expression of PRL and PRLR in ovarian cancer. The current paper advances our knowledge regarding the HCMV related cancers and the possible role of HCMV in these types of cancers. All the experiments and their results were convincing.
Major experiment
The authors are requested to perform IHC on the ovarian cancer tissues with HCMV specific and PLR specific antibodies to see HCMV pathogenesis in the ovarian cancer tissues. This experiment will explain the high PLR expression in the HCMV infected cells as compared to the uninfected cells.
Author Response
Response to Reviewer 1 comments:
We thank the reviewer for the valuable comments, which we think definitely improved the quality of our manuscript. We have addressed the comment below in the point-by-point response and rewritten the manuscript accordingly
Point-by-point response:
Point 1: The authors are requested to perform IHC on the ovarian cancer tissues with HCMV specific and PLR specific antibodies to see HCMV pathogenesis in the ovarian cancer tissues. This experiment will explain the high PLR expression in the HCMV infected cells as compared to the uninfected cells.
Response : We thank the reviewer for this comment. We agree that it is very relevant to use double staining IHC to analyse PRLR and HCMV in ovarian cancer sections. Our investigation using IHC for PRLR and HCMVs different proteins (IE and pp65) stained in the ovarian cancer tissues shows that the majority of HCMV positive cells within the tumor tissues are expressing PRLR. However, PRLR expressing cells within the tumor tissues are not always positive for HCMV proteins, which indicate both direct, and possibly also indirect effects of HCMV on PRLR expression. Due to the access to limited number of ovarian cancer tissues in the present study, we could not proceed with double staining of both HCMV proteins and PRLR. This information is now included in the revised manuscript in results section.
Reviewer 2 Report
Rahbar et al. in this clearly written manuscript embarked on investigating a potential effect of human cytomegalovirus (HCMV) infection on the prolactin system in ovarian cancer cells and tissue. The scope of this manuscript might be interesting for a broader audience as the described studies encompass aspects of cancer biology, virology and cell signaling. The manuscript describes novel association studies for a possible regulation of the prolactin system in HCMV-infected ovarian cancer cells that might be important for an oncomodulatory potential of the virus.
A high serum level of a growth factor prolactin (PRL) is postulated to be a biomarker for ovarian cancer patients. The expression of prolactin receptor (PRLR) and HCMV proteins are often observed in ovarian tumors. Moreover, activation of the PRL/PRLR signaling pathways is linked to cancer via activation of PI3K, AKT and MAPK pathways. Those pathways are also known to be activated by HCMV infection.
In this study, Rahbar et al. analyzed the expression of PRL and PRLR in HCMV-infected ovarian cancer cells (SKOV3) as well as monitored PRLR and HCMV protein levels in ovarian tumor tissue specimens.
Rahbar et al. demonstrated that PRL and PRLR transcript levels were gradually increased in HCMV-infected SKOV3 cells up to 5 dpi and that PRL treatment stimulated the PRLR expression in uninfected cells, but didn`t seem to have any effect on already elevated PRLR in HCMV-infected cells. Additionally, by investigating protein expression the authors presented data suggesting a correlation between HCMV protein expression and heightened levels of PRL and PRLR in SKOV3 cells. By having a unique access to ovarian tumor tissue specimens, the authors were able to document an association in this tissue between HCMV protein and PRLR expression.
The presented results are convincing and support authors’ conclusions, however there are a few aspects of the manuscripts that require clarification and a couple of simple experiments are recommended t strengthen conclusions of this manuscript.
Major comments:
A half of the manuscript is based on the use of SKOV3 cells as a model for HCMV infection in ovarian cancer cells. MOI of 5 is used to infect SKOV3 cells, however no information is provided about the efficiency of infection. Cancer cells are known to be notoriously difficult to infect with HCMV, therefore an information describing % infected cells as well as characteristics of HCMV infection in SKOV3 cells (productive vs. non-productive infection) would be very helpful in a further interpretation of presented results. Especially, it is important in evaluating experiments using the PRL treatment, as PRL may have a distinct effect on uninfected and infected cells. In Fig. 1 and 2, the PRL treatment was used before and at the time of infection and its effect on PRL and PRLR expression was monitored. The authors should explain the hypothesis that drove their decision to use the PRL treatment. The authors should also comment why the PRL treatment led to elevated PRLR mRNA (Fig. 2C), but not PRLR protein (Fig. 3) in uninfected cells. Examining the effect of viral entry vs. viral protein expression vs. a broad cellular stress as triggers of PRL/PRLR expression will significantly enhanced authors’ conclusions. It could be done experimentally by using UV-irradiated virus and gancyclovir with a possible siRNA treatment to HCMV IE, as previously done by this research group. In Fig. 4, instead of omitting primary antibody staining, a HCMV negative tissue specimen will be a proper control that will allow an assessment of the correlation between HCMV infection PRLR expression. The authors only state p<0.05 for the statistical significance. Is that the lowest p value calculated? If not p<0.01 and p<0.001 should be noted in figure legends and in figures.
Minor comments:
In Introduction, it would be beneficial for the manuscript to mention published connection between cyclophilin A and prolactin receptor (Syed et al.), which together with a known importance of cyclophilin A pathway in HCMV infection (i.e. Abdullah et al.) may provide an additional topic for a discussion. Lanes 115 and 118 – sentence redundancy; Figure 1 - Panel B should become C and panel C should become B to correspond with the presentation of results in the text; Figure 3 – densitometry will be very informative and help interpreting the results; Lane 257 – error in the name of RNA isolation kit; The primer and probe sequences should be stated; Lane 280 – error in a name of a buffer; Lane 387, 395, 444 – formatting errors in References; In figure legends, a word “detected” should be changed to monitored/assayed to describe the factual experimental design.
Author Response
Response to Reviewer 2 comments :
We thank the reviewer for the valuable comments, which we think improved the quality of our manuscript. We have addressed the comment below in the point-by-point response and rewritten the manuscript accordingly.
Point by point response :
Major comments:
A half of the manuscript is based on the use of SKOV3 cells as a model for HCMV infection in ovarian cancer cells. MOI of 5 is used to infect SKOV3 cells, however no information is provided about the efficiency of infection. Cancer cells are known to be notoriously difficult to infect with HCMV, therefore an information describing % infected cells as well as characteristics of HCMV infection in SKOV3 cells (productive vs. non-productive infection) would be very helpful in a further interpretation of presented results. Especially, it is important in evaluating experiments using the PRL treatment, as PRL may have a distinct effect on uninfected and infected cells. In Fig. 1 and 2, the PRL treatment was used before and at the time of infection and its effect on PRL and PRLR expression was monitored. The authors should explain the hypothesis that drove their decision to use the PRL treatment. The authors should also comment why the PRL treatment led to elevated PRLR mRNA (Fig. 2C), but not PRLR protein (Fig. 3) in uninfected cells. Examining the effect of viral entry vs. viral protein expression vs. a broad cellular stress as triggers of PRL/PRLR expression will significantly enhanced authors’ conclusions. It could be done experimentally by using UV-irradiated virus and gancyclovir with a possible siRNA treatment to HCMV IE, as previously done by this research group. In Fig. 4, instead of omitting primary antibody staining, a HCMV negative tissue specimen will be a proper control that will allow an assessment of the correlation between HCMV infection PRLR expression. The authors only state p<0.05 for the statistical significance. Is that the lowest p value calculated? If not p<0.01 and p<0.001 should be noted in figure legends and in figures.
We thank the reviewer for very insightful comments.
Point 1: Cancer cells are known to be notoriously difficult to infect with HCMV, therefore an information describing % infected cells as well as characteristics of HCMV infection in SKOV3 cells (productive vs. non-productive infection:
Response :
-We have now included the information regarding efficiency of HCMV infection at different dpi in the revised manuscript in Results section.
-To our knowledge major efforts have been done to try to isolate HCMV from in vivo infected cancer cells but so far no one has managed to recover infectious virus from primary tumours. Few examples exist of productive HCMV infection in cancer cells infected in vitro by laboratory-adapted strains (T98G glioblastoma cells are semi permissive for HCMV infection). In our hands, we so far did not succeed in recovering infections virus from cancer cells infected in vitro. This is a challenge in the HCMV cancer field that will hopefully be solved in the near future.
Point 2: The authors should explain the hypothesis that drove their decision to use the PRL treatment.
Response :We have studied the impact of HCMV on cancer cells for many years, and awoke an interest in PRL/PRLR system due to collaboration between the two senior authors on this manuscript. The idea of using PRL treatment in uninfected and HCMV infected cells starting before or during infection was to design a study resembling two scenarios in an in vivo The results would give guidance to explain the different levels of PRL and PRLR in a situation in vivo when PRL is already induced before infection or is increased at the time of infection in the ovarian cancer patients. This information has been included in the revised version of the manuscript in the Material and Methods section.
Point 3: The authors should also comment why the PRL treatment led to elevated PRLR mRNA (Fig. 2C), but not PRLR protein (Fig. 3) in uninfected cells.
Response: Our hypothesis related to this finding is that PRL treatment of the SKOV3 cells leads to increased PRLR transcript and protein production. However, the high levels of PRLR transcripts but low levels of proteins during PRL treatment may be explained by an inhibitory post translational mechanism to control PRLR protein translation in PRL treated cells. This information has been included in revised version of the manuscript in the Discussion section.
Point 4: It could be done experimentally by using UV-irradiated virus and gancyclovir with a possible siRNA treatment to HCMV IE, as previously done by this research group.
Response :We are agreement with the reviewer that an experiment using UV-irradiated virus and ganciclovir treatment or siRNA treatment to control HCMV IE expression, would add more valuable information about our findings and clarify if replication of HCMV or only viral entry and production of different immunological factors results in increased levels of PRL and PRLR. However, this experiment is time consuming and will take several weeks to complete, and we did not receive enough extension time from the journal to perform these experiments. We respect the editorial decision on this matter, and therefore consider these experiments beyond the scope of the current study. In our continued studies on this concept, we will prioritize these experiments.
Point 5: In Fig. 4, instead of omitting primary antibody staining, a HCMV negative tissue specimen will be a proper control that will allow an assessment of the correlation between HCMV infection PRLR expressions.
Response : We have now included IHC on HCMV negative and weakly PRLR positive human placenta tissue as figure 4C.
Point 6: The authors only state p<0.05 for the statistical significance. Is that the lowest p value calculated? If not p<0.01 and p<0.001 should be noted in figure legends and in figures.
Response :The lowest p value is calculated and a value of p<0.05 is considered statistically significant. This information has been included in the revised manuscript in section Statistical Analysis.
Minor comments:
Point 1: In Introduction, it would be beneficial for the manuscript to mention published connection between cyclophilin A and prolactin receptor (Syed et al.), which together with a known importance of cyclophilin A pathway in HCMV infection (i.e. Abdullah et al.) may provide an additional topic for a discussion.
Response :Thank you for this comment. We agree that cyclophilin A is indeed relevant in this context and we have added the proposed references in the revised version of the manuscript with the recommended references. (Ref number 28 and 29)
Point 2: Lanes 115 and 118 – sentence redundancy;
Response :The redundancy was not found.
Point 3: Figure 1 - Panel B should become C and panel C should become B to correspond with the presentation of results in the text;
Response :This correction has been done, upon the reviewer´s request.
Point 4: Figure 3 – densitometry will be very informative and help interpreting the results;
Response: Densitometry graphs are now included in the revised version of the manuscript and the results are written both in the figure legend and in result section 2.3.
Point 5: Lane 257 – error in the name of RNA isolation kit; the primer and probe sequences should be stated;
Response :This information has been included in revised version of the manuscript in the Material and Methods section.
Point 6: Lane 280 – error in a name of a buffer
Response :Trish changed to Tris
Point 7: Lane 387, 395, 444 – formatting errors in References;
Response: Formatting errors correction had been done to references specified.
Point 8:In figure legends, a word “detected” should be changed to monitored/assayed to describe the factual experimental design.
Response :In the figure legend the word detected has been changed to monitored as suggested.
This manuscript is a resubmission of an earlier submission. The following is a list of the peer review reports and author responses from that submission.